# Stokes Dynamic Polarimeter for Non-Organic and Organic Samples Characterization

**DOI:** 10.3390/s22062155

**Published:** 2022-03-10

**Authors:** Dora-Luz Almanza-Ojeda, Daniela Rodriguez-Sotelo, Rogelio Castro-Sanchez, Rene Martinez-Celorio, Mario-Alberto Ibarra-Manzano

**Affiliations:** 1Department of Electronics Engineering, Universidad de Guanajuato, Salamanca 36885, Mexico; dora.almanza@ugto.mx (D.-L.A.-O.); daniela.rodriguez@ugto.mx (D.R.-S.); ibarram@ugto.mx (M.-A.I.-M.); 2Physics Department, Miami Dade College, Miami, FL 33132, USA; rmarti23@mdc.edu

**Keywords:** light polarization, Mueller matrix, photoelastic modulator, synchronization, surface fruit

## Abstract

The light polarization properties provide relevant information about linear–optical media quality and condition. The Stokes–Mueller formalism is commonly used to represent the polarization properties of the incident light over sample tests. Currently, different Stokes Polarimeters are mainly defined by resolution, acquisition rate, and light to carry out accurate and fast measurements. This work presents the implementation of an automatic Stokes dynamic polarimeter to characterize non-biological and biological material samples. The proposed system is configured to work in the He-Ne laser beam’s reflection or transmission mode to calculate the Mueller matrix. The instrumentation stage includes two asynchronous photoelastic modulators, two nano-stepper motors, and an acquisition data card at 2% of accuracy. The Mueller matrix is numerically calculated by software using the 36 measures method without requiring image processing. Experiments show the efficiency of the proposed optical array to calculate the Mueller matrix in reflection and transmission mode for different samples. The mean squared error is calculated for each element of the obtained matrix using referenced values of the air and a mirror. A comparison with similar works in the literature validates the proposed optical array.

## 1. Introduction

The design of biosensors has recently shown a considerable advance in human diagnosis through proteins and enzymes detection that characterize bacteria and virus diseases [1]. In detecting specific viruses such as SARS-CoV 2 [2] and more biomedical and clinical diagnoses [3], some of these biosensors use the polarization of light as a detection technique. The study of polarized light and methods of polarization measurements have increased due to their variety and vast applications. Polarimetry techniques allow for the identification of properties of surfaces by analyzing the changes in the polarized light [4]. Some applications of polarized light include the characterization of microstructures in biological tissues [5], the prevention and detection of illnesses like cancer or cirrhosis [6,7], materials classification [8,9,10], and analysis of components on industrial products, among many other biomedical and industrial areas. The Mueller matrix is used to describe the interaction between light and different optical means to develop potential methods of diagnosis in biomedical science [11,12]. The Mueller matrix is the numerical representation of the polarization states of incident light on the sample surface. Thus, it plays a vital role in characterizing samples’ structural properties associated with their Stokes Vectors. 

The Stokes–Mueller theory [13] includes 36 measurements representing the different states of the polarization of light. The conventional optical array comprises devices such as polarizers and retarder waveplates. Usually, instrument calibration based on polarized light [14] is carried out by calculating the Mueller matrix for one or several optical elements, considered as reference, and shown in the state-of-the-art [15,16,17]. Once the polarimeter is calibrated, the samples under test are measured. 

A polarimeter is considered dynamic when its design includes photoelastic modulators. These modulators generate output signals based on the birefringence properties [18] produced by the SiO2 piece located at the electronic circuit’s output. The birefringence properties change in function of the input voltage of the modulator, generating linear and circular polarization states in the output signal. Thus, the dynamic modulation range of the polarimeter works with a random and polarized light source that passes through the optical array composed of polarizers and photoelastic modulators. Usually, this assembly can be used in transmission or reflection mode for different material samples.

This paper presents the design and implementation of a dynamic polarimeter in transmission and reflection mode for calculating the Mueller matrix of different samples. Electronic components are chosen to provide high accuracy and speed during dynamic measurements. The advantage of the proposed system is to compute the Mueller matrix using the 36 measurements method without using image processing. The instrumented array automatically acquired the measurements to calculate the numerical coefficients of the matrix. The matrices are obtained using a Graphical Unit Interface based on LabView of National Instruments^®^. The proposed polarimeter is calibrated by calculating the Mueller matrices of the air and a mirror and comparing with reference values presented in state-of-the-art. Then, preliminary apple and banana matrices are proposed for establishing a quantitative reference of the polarization properties of fruit tissues. To the best of our knowledge, Mueller matrices have not been submitted to describe the behavior of the polarized light through the fruit surfaces. However, the main contribution of our system is the instrumentation and control device to automatically perform the polarization states set up, which allows an economical, accurate, and easy-to-use polarimeter. The proposed electronic components can be seen as modules that can be included in the whole optical array and the PC for matrices calculation by software. Overall system performance is validated experimentally, achieving high values of repeatability and precision.

The rest of the document explains the design and experimental setup of the proposed system. The theoretical concepts of polarization properties of light are defined in Section 2. Section 3 describes the optical array to implement the transmission and reflection modes of the system. Section 4 shows the experimental results using different types of medium and compares them with similar findings. Conclusions and perspectives of the proposed work are presented at the end of this document.

## 2. Background

The Mueller matrix plays an essential role in characterizing the material samples’ microstructural and physical properties. According to the polarization states, different interactions occur between the incident light and the material sample. The individual elements of the matrix register the incident beam changes during the light-sample interaction. Hence, calculating such numerical elements in the matrix requires the study of essential concepts about the polarized light generation and the Stokes formalism, which will be briefly presented in the following subsections.

### 2.1. Polarized Light Generator

A Photoelastic Modulator (PEM) and a polarizer could be time-adjusted by software to generate different polarization states of a reflected beam of light. The variations on the reflected beam show a periodicity on the time domain that allows proposing an array to measure the polarization states dynamically. The basic idea of the PEM is to modulate polarized light [18] that passes through its optical header when the amplitude of the applied periodic voltage varies. This amplitude variation modifies the property of optical birefringence. The optical header of PEM is a device that generates a programmed phase-retarded between the orthogonal components of the electric field. Thus, being the light a transversal electromagnetic wave, the beam’s polarization state allows modulating periodically in the time domain the polarization state of the light beam. PEM is used in multiple experimental methods [19,20], showing essential features such as high sensibility, broad spectral range, and high precision in phase modulation under the Stokes-Mueller formalism.

### 2.2. Stokes-Mueller Formalism

Recently, light polarimetry contributed to analyzing and diagnosing biological tissues, revealing essential information about healthy or pathological tissue status [21]. The structural features in organic or non-organic samples alter the incident light in the sample surface. The changes produced on the incident light after interacting with a material sample can be described by the elements of the Mueller matrix. These elements represent the coefficients of the incident and output Stokes vector. Figure 1 depicts the Stokes–Mueller formalism. An input Stokes vector **S***_i_* affected by a linear system with optical features given by M provides the output Stokes vector **S***_o_*. M is the Mueller matrix that models the optical properties of any material sample under test.

The mathematical representation of Stokes vectors is defined by Equation (1) [16]:**S***_o_* = **M****S***_i_*(1)
where **M** is the Mueller matrix, **S***_x_* represents the lineal polarization states of incident light (*x* = *i*) and reflected light (*x* = *o*) into the optical media. Linear polarization states of light are composed of: linear horizontal (*h*), linear vertical (*v*), linear to 45°(+), linear to 135°(−), and circular polarization states, right circular (*r*) and left circular (*l*). Combining these states helps determine variations on physical structures of material samples during polarized light incidence.

The Mueller matrix of a lineal optical media is a 4 × 4 matrix whose elements provide the media’s anisotropy information [22]. In this work, the Mueller matrix is calculated using the 36 measurements method [23]. Thus, six polarization states are generated from the incident source of light (**S***_i_*); six reflected light (**S***_o_*) states are analyzed. This method is mainly used to measure unknown optical properties of samples under tests (arbitrary physical systems). Equation (2) illustrates Mueller matrix elements based on the irradiances *I*. Each element *m*_ab_ of the matrix depends on two irradiances *I_xy_*. The subscripts at each member of M represent the polarized input states established by the incident Stokes vector **S***_i_* and the polarized output states corresponding to the reflected Stokes vector **S***_o_* [22].
*m*_11_ = ½ (*I*_*h**h*_ + *I*_*h**v*_ + *I*_*v**h*_ + *I*_*v**v*_)      *m*_12_ = ½ (*I*_*h**h*_ + *I*_*h**v*_ − *I*_*v**h*_ − *I*_*v**v*_)
*m*_13_ = ½ (*I*_+*h*_ + *I*_+*v*_ − *I*_−*h*_ − *I*_−*v*_)      *m*_14_ = ½ (*I*_*r**h*_ + *I*_*r**v*_ − *I*_*l**h*_ − *I*_*l**v*_)
*m*_21_ = ½ (*I*_*h**h*_ − *I*_*h**v*_ + *I*_*v**h*_ − *I*_*v**v*_)      *m*_22_ = ½ (*I*_*h**h*_ − *I*_*h**v*_ − *I*_*v**h*_ + *I*_*v**v*_)
*m*_23_ = ½ (*I*_+*h*_ − *I*_+*v*_ − *I*_−*h*_ + *I*_−*v*_)      *m*_24_ = ½ (*I*_*r**h*_ − *I*_*r**v*_ − *I*_*l**h*_ + *I*_*l**v*_)
*m*_31_ = ½ (*I*_*h*+_ − *I*_*h*−_ + *I*_*v*+_ − *I*_*v*−_)      *m*_32_ = ½ (*I*_*h*+_ − *I*_*h*−_ − *I*_*v*+_ + *I*_*v*−_)
*m*_33_ = ½ (*I*_++_ − *I*_+−_ − *I*_−+_ + *I*_−−_)      *m*_34_ = ½ (*I*_*r*+_ − *I*_*r*−_ − *I*_*l*+_ + *I*_*l*−_)
*m*_41_ = ½ (*I*_*h**r*_ − *I*_*h**l*_ − *I*_*v**l*_ + *I*_*v**r*_)      *m*_42_ = ½ (*I*_*h**r*_ − *I*_*h**l*_ − *I*_*v**r*_ + *I*_*v**l*_)
*m*_43_ = ½ (*I*_+*r*_ − *I*_+*l*_ − *I*_−*r*_ + *I*_−*l*_)      *m*_44_ = ½ (*I*_*r**r*_ − *I*_*r**l*_ − *I*_*l**r*_ + *I*_*l**l*_)(2)

## 3. Materials and Methods

### 3.1. Experimental Setup

The schematic diagram of the proposed dynamic polarimeter in reflection mode is shown in Figure 2. The dynamic polarimeter is an optical assembly configured in reflection mode that consists of five parts: a light source module (laser), a polarization state generator (PSG), a sample under test, a polarization state analyzer (PSA), and a photodetector (PD).

Hence, the first array adopted was configured in reflection mode composed of a 632 nm He-Ne Laser with power output at 17 mW in random polarization. The light beam passes through the polarization state generator (PSG) to generate the six polarization states. The PSG consists of a linear polarizer with a transmission axis fixed at +45° to the horizontal axis; additionally, it includes an optical head PEM100 Photoelastic modulator (Hinds Instruments, OR, USA, [18]) with a transmission axis aligned to 0° with the horizontal axis. After that, polarized light is transmitted or reflected through a sample, which could also be a material substance or the air. Next, the polarization state Analyzer (PSA) consists of an optical array to analyze six polarization states of the light. This PSA includes a Photoelastic modulator (PEM100, Hinds Instruments^®^, OR, USA), with a transmission axis aligned to 0° with the horizontal axis and a transmission axis oriented to −45° with the horizontal. Both polarizers, PSG and PSA, use two Stepper motors (NR360S, Thorlabs^®^, Newton, NJ, USA) to generate the combination of circular and linear polarization states. Finally, the proposed assembly includes a switchable gain photodetector PD (PDA36A, Thorlabs^®^, Newton, NJ, USA) to measure the modulated polarized light.

As mentioned above, the main contribution of the proposed optical assembly is the generation of the circular and linear polarization states autonomously, combining static and dynamic methods and employing an embedded synchronization circuit. The six polarization states are generated in the PSG block and six more in the PSA block. The combination among these polarization states generates the 36 measurements in the photodetector to determine the Mueller matrix. From these 36 measurements of the polarization states, 16 correspond to linear states, and 20 to linear and circular states. The overall methodology consists of four stages: (1) alignment, (2) generation of linear polarization states, (3) generation of circular polarization states, and (4) synchronization. These four stages were also instrumented in transmission mode for this study.

### 3.2. Alignment

The calibration of the proposed polarimeter requires that each component of the experimental assembly be aligned. Initially, the alignment is performed for transmission mode by setting the light source, the PSG, the sample, the PSA, and the photodetector in the same line as illustrated in Figure 3. A sequence of pulses emitted by software allows extracting minimum, mean, maximal, and normalized values for the air. After that, the 36 measures are automatically computed on the software interface (LabVIEW) to obtain the Mueller matrix of the air as a reference. The matrix of the air shows a positive “1” value on its diagonal elements [24]. Hence, the PSG or PSA must be realigned when diagonal values differ to “1”.

As illustrated in Figure 2, the light source module is oriented at 30° in the PSA and photodetector directions in reflection mode. This configuration setup uses a front surface mirror as a reference sample. Like transmission mode, 36 measurements are automatically calculated using the graphical user interface on LabVIEW to determine the Mueller matrix parameters. Ideally, the diagonal elements *m*_11_ and *m*_22_ in the Mueller matrix of the mirror are close to 1. Unlike, *m*_33_ and *m*_44_ are close to −1. The diagonal elements with low similarity in the ideal values indicate that the PSG must be realigned.

### 3.3. Combination of Linear Polarization States

The generation and analysis of the linear polarization states require the PSG and PSA stages. Each polarizer is mounted on an N360S motorized rotation stage of Thorlabs^®^. A control system programs both motors to set the transmission axis for each polarizer in four specific positions. The 16 combinations required for input and output are calculated from these positions. The data card model USB-6259 distributed by National Instruments^®^ acquires the signals of each polarization state to the PC and generates the synchronized signals to control the instrumented array. Thus, the modules are automated using LabVIEW^®^. Figure 4 shows the flow diagram programmed in a subVI of LabView software. This strategy implements the proposed dynamic polarimeter that generates and controls various position combinations for each stepper motor.

Linearly polarized light generates polarization by setting off the PEM optical headers but keeping on the data acquisition process. Table 1 shows the combinations performed by our optical array system in transmission mode to determine the Mueller matrix for different linear-optical media. Initially, the reference point of the stepper motors is set to 0°. The first column of Table 1 shows the combination of linear polarization states used for passing a laser beam through the transmission axis of the polarizers. The second column shows each position of the stepper motor. The third and fourth columns illustrate the theoretical and normalized expected values.

Several measurements had been acquired to increase the number of samples taken from the polarization states of Table 1. These measurements are averaged for each pair of polarization states during a fixed time interval. The measurements are stored in a file to calculate the Mueller matrix elements shown in Equation (2).

#### Linear and Circular Polarization States Combinations

Figure 2 and Figure 3 show the schematic representation of the instrumented optical array used to generate linear and circular polarization states. Table 2 illustrates the 20 linear and 16 circular polarizations states to determine the Mueller matrix. The first column of Table 2 shows the states generated and analyzed. Linear polarizers PL1 and PL2 are presented in the second and fifth columns. The third and sixth columns illustrate the combined on/off conditions for the PEM1 and PEM2. The fourth and seventh columns show retarder waveplates values (λ) introduced to generate some circular polarization states.

Using Table 2, linear and circular polarization states are generated by combining different angles and states of the polarimeters. This experiment is similar to the states generated using the PSG and PSA blocks in reflection mode shown in Figure 2. For instance, in the four first states (−*l*, −*r*), the state “−“ (represented by a minus sign) is performed by setting to 135° the PL1 polarizer in the PSG block and setting the PEM1 to off. A similar combination is performed for the “*l*” state, but in this case, the PL2 polarizer in the PSA block is set to 135°. The PEM2 is also on and delayed by λ/4. The two states (−*r*) are generated and analyzed using similar values for PSG and PSA. The states *l* and *r* are dynamically generated at different times, while PEM2 is set to *on* and delayed by λ/4 programmed on LabVIEW. Similarly, the rest of the states proposed in Table 2 are generated. In the last row of Table 2, for the states (*rr*, *ll*, *rl*, *lr*), both PEM1 and PEM2 are turned on and synchronized [25]; furthermore, PL1 and PL2 polarizers are set to 45° and −45° to the horizontal, respectively [26].

### 3.4. Synchronization

The dynamic polarimeter proposed uses two optical heads (Hinds Instruments) to integrate the PEM modules. In practice, we found a difference of around 6 Hz in the oscillation frequencies in both devices. The measurements registered from PEM modules must be synchronized to provide correct data values. A phase detector is included to synchronize both PEM modules with a reference point. To set this reference point, we measure the phase difference between both oscillation frequencies oriented at 0° and 180°. For that purpose, we take advantage of each PEM that provides a TTL output signal equivalent to the input signal frequency that passes through the PEMs. Then, a digital phase detector composed of a type II phase comparator is used to detect TTL digital output and the frequency differences to achieve the synchronization PEMs [27].

Figure 5 shows the graphical output signal of the phase-cero detector circuit monitored by an oscilloscope to demonstrate the functionality of the synchronization stage. This circuit provides an output digital pulse ④ when the phase difference is 0° between ① and ② input signals. This pulse is used as a reference if periodic polarization states in the PEMs are detected. Using PEMs synchronization parameters, it is possible to obtain 20 measurements in specific points related to the combination of linear and circular polarization states. In our case, the reference point is the transition from low to high level at the output of the phase detector (see channel ④ in the oscilloscope).

Figure 6 illustrates the use of the phase detector to synchronize the NI-USB acquisition card and the oscilloscope during the detection of the polarization states.

The polarization states *r* and *l* in the signal propagated through the optical header of each PEM are specified in the datasheet device [18]. Figure 7 illustrates the PEMs monitored by an oscilloscope. Two dynamic polarization states are measured using the acquisition data card at the output of the zero-phase detector. Figure 7a displays the oscilloscope screen for *r* and *l* polarization states, and Figure 7b shows the *l*,*l* states.

Once the system is referenced to a specific point in time established for the synchronization process, we acquire various measurements automatically at this reference point and calculate the mean values of the instantaneous measurements for obtaining a reliable value.

## 4. Results

Experimental tests validate the performance and accuracy of our dynamic polarimeter assembly. In the experiments, the room temperature was between 20 °C and 25 °C, and the laser was pre-heated for 30 min before use. For each set of measurements, the overall system is calibrated by software using frequency differences between the PEMs. To obtain the differences, we average the location of maximum and minimum values measured at the output of the photodetector. These values could vary depending on the selected gain in the photodetector; in these experimental tests, we used a fixed gain of 20 db. The retarded output signal pulse measurements in the phase detector showed a periodicity of 100 ns. The air and a mirror are used as standard samples in transmission and reflection mode, respectively, to verify the system calibration. Figure 8 shows the proposed optical system implemented in reflection mode for two different degrees of orientation, 30° and 165°.

Each Mueller matrix is compared with its corresponding ground-truth matrix proposed in the literature [22]. The difference between the proposed matrix and the ideal matrix established for standard samples is calculated using the mean square error metric (see Equation (3)). The mean error obtained indicates the deviation of matrix elements to their ideal values but in general. The further numerical analysis allows the determination of the specific details in the matrix with maximal differences.
(3)$$MSE=\frac{1}{16}\overset{4}{\underset{i=1}{\sum }}\overset{4}{\underset{j=1}{\sum }}\left|M_{ij}^{ideal}-M_{ij}\right|$$

The following subsections describe the Mueller matrix measured by our proposed dynamic polarimeter and compare the matrix elements that show the higher accuracy error. The comparison results demonstrate the reliability of the instrumented system capable of measuring linear and circular polarization effects of some organic and non-organic samples without using CCD sensors. Additional methods, like Mueller matrix transformation or decomposition, provide a set of microstructural properties of most biological tissues; however, the comparison and computation of such properties are reserved for future works.

### 4.1. Mueller Matrix in Transmission Mode

The Mueller matrix in transmission mode measures two samples: the air and a linear polarizer with the transmission axis perpendicular to the horizontal. The first row of Table 3 shows our proposed Mueller matrix for the air [16]. According to this reference matrix, the diagonal elements are one or close to one, indicating that our system is calibrated correctly. The mean error indicates 2.12%, being *m*_13_, *m*_23_, and *m*_32_ the elements with maximal error. The remaining elements of the matrix show a minimal accuracy error regarding ideal values. This comparison indicates that such elements must be considered the most critical during matrix calibration due to the higher accuracy errors.

The second row in Table 3 shows the Mueller matrix of a polarizer with the transmission axis set to 90° to the horizontal. The elements $m_{11}$, $m_{12}$, $m_{21}$, and $m_{22}$ in this matrix are close to 1, similar to ideal values proposed in the literature for a linear polarizer [28]. The mean error obtained for this matrix is 2.23%, being $m_{13}$, $m_{23}$, $m_{31}$, and $m_{32}$ the elements with maximal error.

### 4.2. Mueller Matrix in Reflection Mode

A typical material used for calculating Mueller matrices is a front mirror surface. The surface of this mirror perfectly reflects a beam of light like a half-wave plate [4,25,26]. As a reference, we calculate the matrix for a front mirror surface in reflection mode, yielding an error of 0.31% compared with previous work (see Table 4). For a mirror, $m_{33}$ and $m_{44}$ diagonal elements show a minus sign [29], validating the assumption that a front mirror surface is perfectly reflected, like a retarder half-wave plate.

Before testing with organic samples, similar experimental tests were performed with the dynamic polarimeter in reflection mode using different reflected surfaces. Thus, we replaced the front mirror surface with a polarizer with the transmission axis oriented at 90° and a rough aluminum surface. The experimental results are shown in Table 5. The Mueller matrices for both elements are not provided in the literature to the best of our knowledge. Hence, we have included both matrices without an MSE score. Note that diagonal elements in both matrices show similar behavior to matrix elements of a mirror; the first two elements are positive and the last ones negative. The experimental array mounted for measuring rough aluminum surface is illustrated in Figure 9.

### 4.3. Mueller Matrix for Fruit Surfaces

The dynamic polarimeter is validated using different organic samples in reflection modes, such as an apple and a banana. Table 6 illustrates the corresponding Mueller matrices calculated in reflection mode.

Figure 10 and Figure 11 show the physical experimental array implemented to measure the Mueller matrices of an apple and a banana, respectively. The $m_{11}$, $m_{22}$, $m_{23}$, and $m_{33}$ elements of the matrix are sensitive to linear polarizing light beams. The $m_{44}$ element represents the reaction to circular polarization. We note that this element was significantly reduced to the mirror surface, which is −1, indicating circular depolarization of the surface.

The Mueller matrices illustrated in Table 6 are general approximation matrices obtained from an instrumented dynamic polarizer. This is the first time that Mueller matrices for fruits are proposed to the best of our knowledge. We can reduce some errors in the matrix elements associated with the number of tests performed by measuring different apples and bananas from a batch of fruit. These matrices provide much information on the sample surface based on the fruits’ maturity degree.

## 5. Discussion

This work proposes a Stokes polarimeter system that calculates the Mueller matrix of organic and non-organic samples by transmission or reflection. Instrumentation of polarizers and stepper motors allow the 36 measurements method to achieve the same precision for every experimental sample. The main contribution of this method is to provide an instrumented array for calculating Mueller matrices by measuring polarized light behavior on the sample surfaces. This method proposes an alternative to the typical image of the Mueller matrix; instead, we propose numerically calculating each element of the matrix by acquiring measured signals and using the 36 measurements method. Mueller matrix of fundamental elements such as air, polarimeters, and mirror are used to calibrate and validate the proposed system. Specifically, Mueller matrices of different organic samples are calculated (fruit surfaces) and shown on the main screen of the designed software. The mode can be selected by reflection or transmission. Moreover, even if some matrices cannot be compared to similar experimental values, we achieve a minimal MSE score for common elements like air or polarizers.

Overall, the Mueller matrices obtained for different elements provide a reliable calibration for ideal reference samples and show that our instrumented array can be configured and adapted to measure lighting properties of additional organic and non-organic materials. Related works proposed Mueller matrices measured using a CCD camera with higher error accuracy values than our proposed optical array. For instance, in [30], the Mueller matrices of the air or polarizers using a CCD camera show around 4% of error and 2% in our case. In contrast, reference [31] also measures the matrix for the air achieving an error of 0.2% and even lower for some individual elements of the matrix. Both works measure the optical properties of the air in transmission mode but using two different models of CCD cameras, the camera with a higher dynamic range provides notably more accurate results. In our case, we have proposed a photodetector instead of a CCD camera because we can vary the acquisition rate of the photodetector and acquire a high number of samples to calculate the Mueller matrix. Additionally, the photodetector measurements require fewer saving storage and memory resources. Moreover, this device provides high bandwidth for measuring the polarization properties of the samples under tests. So far, the photodetector offers an alternative to the CCD camera commonly used for Mueller matrix determination.

Some additional aspects for improvement in future works are listed: (1) the design of a compact and portable prototype for measuring different kinds of samples under the same conditions using reflection mode; (2) Extend the current wavelength of the incident light to a broader spectrum range for measuring microstructural properties that characterize biological tissues; (3) Compute other Mueller matrix properties for a suitable characterization of material samples in industry or clinical applications.

## Figures and Tables

**Figure 1 sensors-22-02155-f001:**
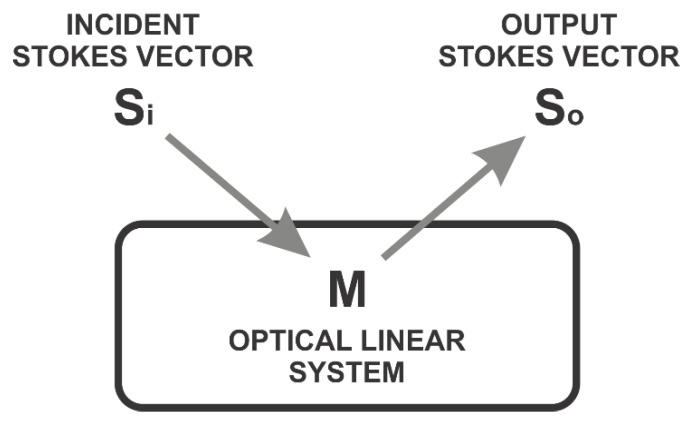
Incident beam light interacting on a lineal optical media.

**Figure 2 sensors-22-02155-f002:**
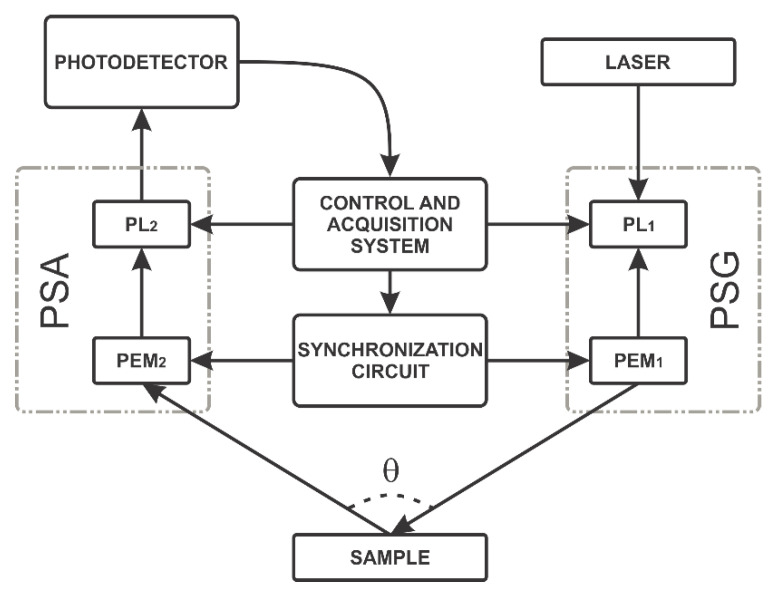
Block diagram of the dynamic polarimeter in reflection mode.

**Figure 3 sensors-22-02155-f003:**
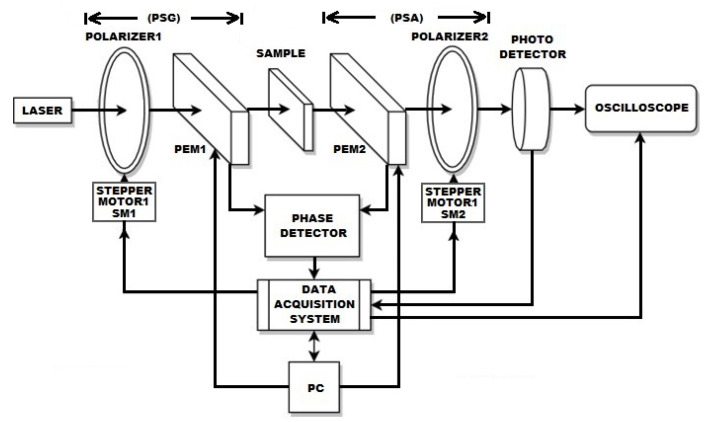
Block diagram of the dynamic polarimeter alignment in transmission mode.

**Figure 4 sensors-22-02155-f004:**
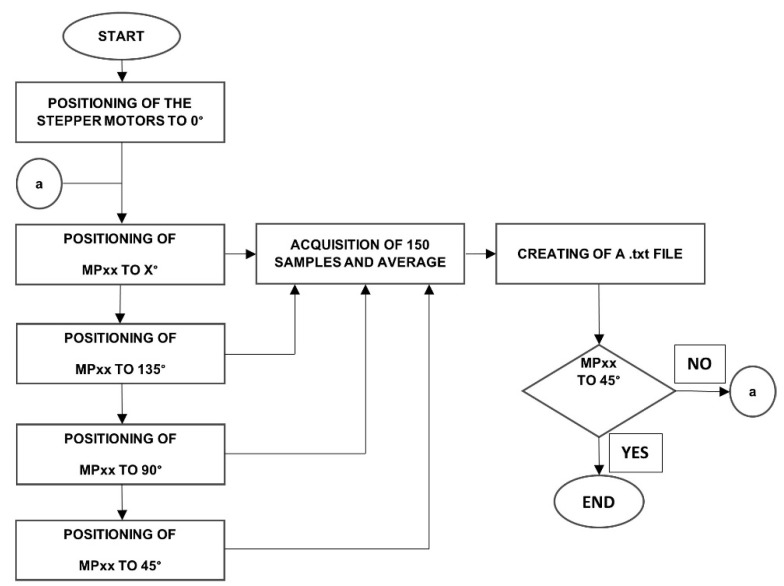
Block diagram of acquisition and states generation of the linear polarimeter.

**Figure 5 sensors-22-02155-f005:**
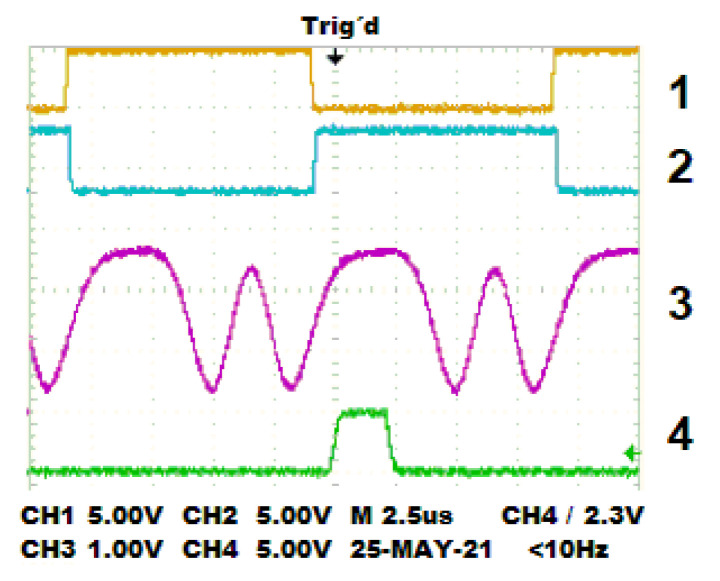
Measured signals: (1) TTL output of PEM1, (2) TTL output of PEM2, (3) output signal of the photodetector, (4) output signal of the phase detector.

**Figure 6 sensors-22-02155-f006:**
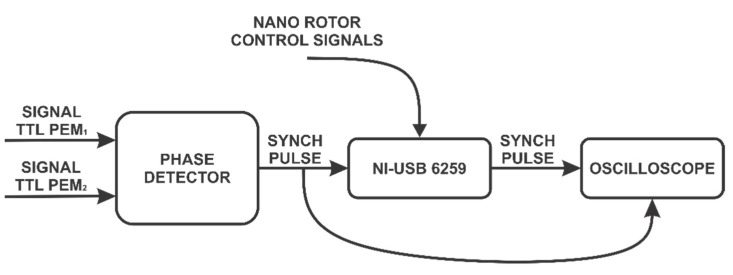
Block diagram of acquisition and states generation of circular polarization.

**Figure 7 sensors-22-02155-f007:**
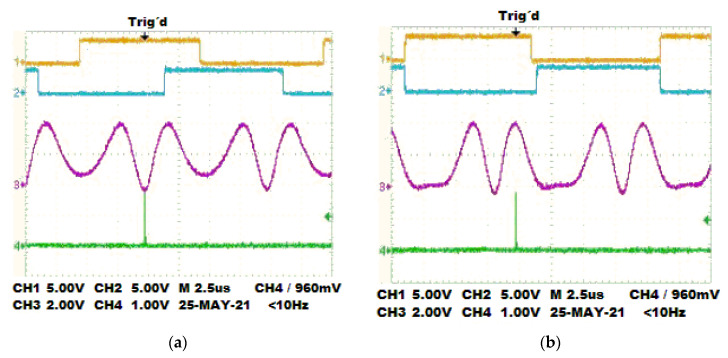
The plot of the polarization states generated. (**a**) Combinations for the *r*,*l* states. (**b**) Combinations for the *l*,*l* states.

**Figure 8 sensors-22-02155-f008:**
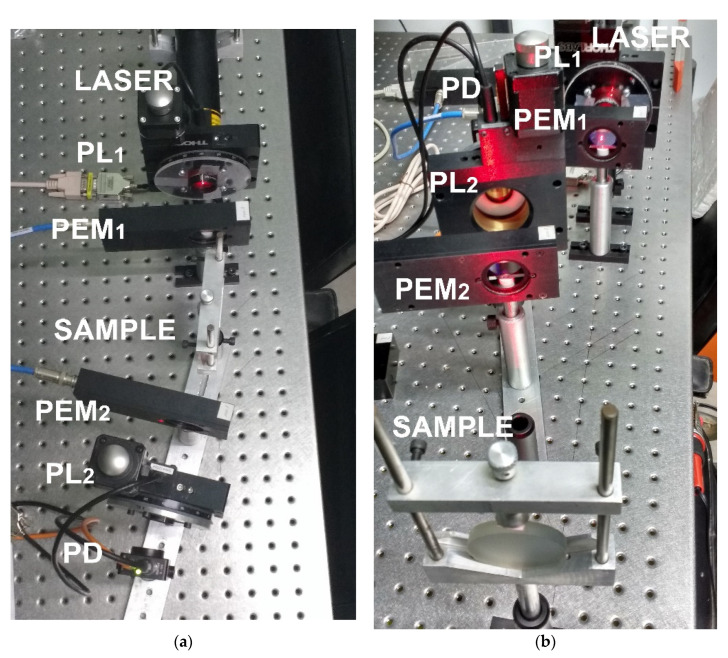
Dynamic Stokes polarimeter in reflection mode: (**a**) oriented at 30°; (**b**) oriented at 165°.

**Figure 9 sensors-22-02155-f009:**
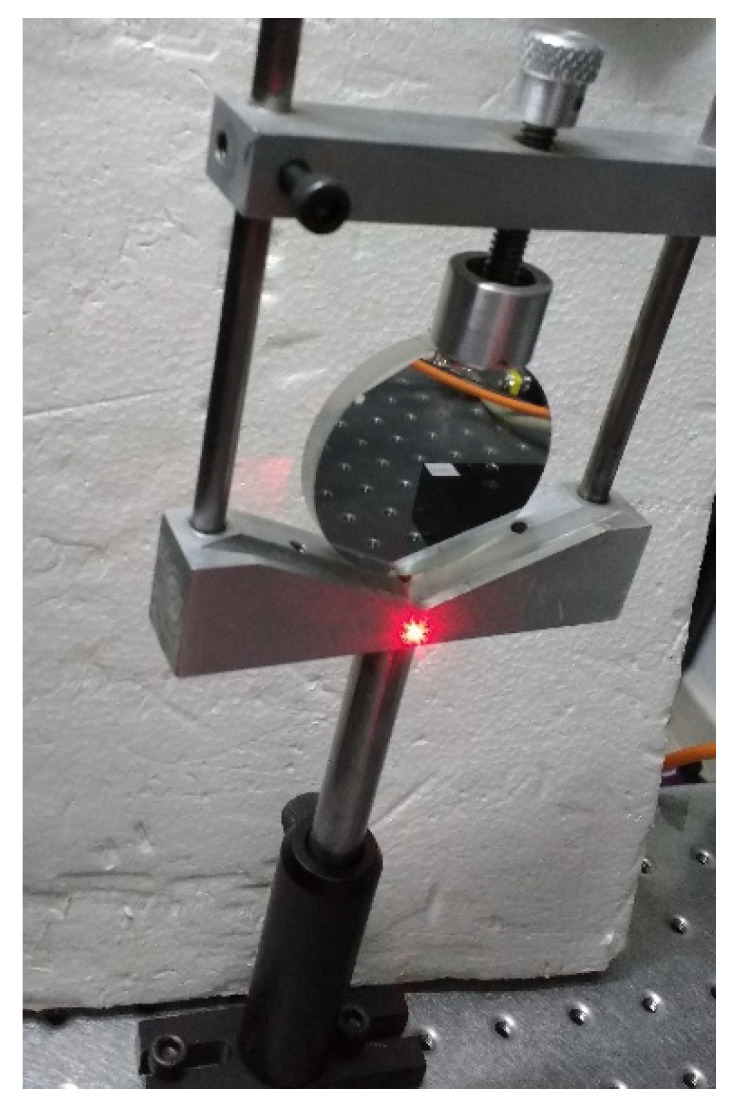
Dynamic Stokes polarimeter in reflection mode through the aluminum surface.

**Figure 10 sensors-22-02155-f010:**
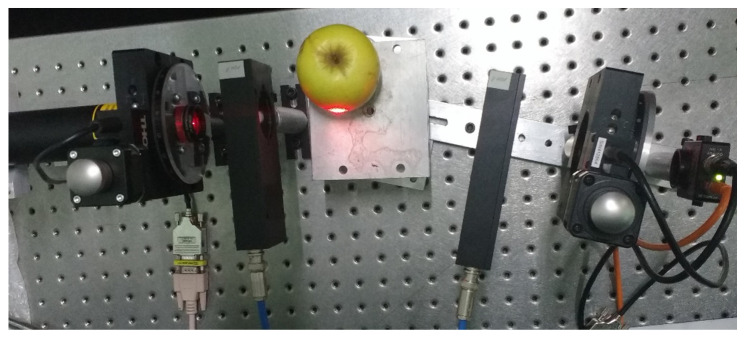
Dynamic Stokes polarimeter in reflection mode through the apple surface.

**Figure 11 sensors-22-02155-f011:**
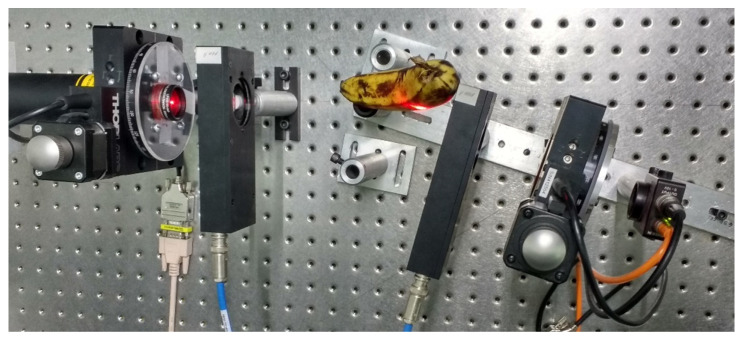
Dynamic Stokes polarimeter in reflection mode through the banana surface.

**Table 1 sensors-22-02155-t001:** Setup parameters to calculate linear polarization states and the normalized irradiance.

State	Angular Position PL1/PL2	Theoretical Irradiance	Normalized Irradiance
*hh*	0°/0°	Maximal	1
++	45°/45°		
*vv*	90°/90°		
−−	135°/135°		
*h*+	0°/45°	Medium	0.5
+*h*	45°/0°		
*h*	0°/135°		
−*h*	135°/0°		
*v*+	90°/45°		
+*v*	45°/90°		
*v*	90°/135°		
−*v*	135°/90°		
*hv*	0°/90°	Minimal	0
*vh*	90°/0°		
*l*−	45°/135°		
−+	135°/45°		

**Table 2 sensors-22-02155-t002:** Parameters to generate linear and circular polarization states.

State	Angle of PL_1_ (PSG)	PEM_1_ Status (PSG)	Λ	Angle of PL_2_ (PSA)	PEM_2_ Status (PSA)	λ
−*l*, −*r*	135°	Off	-	135°	On	λ/4
*l*−, *r*−	45°	On	λ/4	135°	Off	-
*l*+, *r*+	45°	On	λ/4	45°	Off	-
+*l*, +*r*	45°	Off	-	135°	On	λ/4
*hr*, *hl*	0°	Off	-	135°	On	λ/4
*rh*, *lh*	45°	On	λ/4	0°	Off	-
*vr*, *vl*	90°	Off	-	135°	On	λ/4
*rv*, *lv*	45°	On	λ/4	90°	Off	-
*rr*, *ll*, *rl*, *lr*	45°	On	λ/4	135°	On	λ/4

**Table 3 sensors-22-02155-t003:** Mueller matrix for air and a polarizer with the transmission axis set to 90°.

Sample	Experimental Mueller Matrix	Mean Error (e)	Elements with Higher Error
Air	$\left[\begin{array}{cccc} 1.0000 & -0.0203 & -0.0407 & -0.0032\\ -0.0204 & 0.9936 & 0.0677 & -0.0007\\ 0.0819 & -0.0673 & 1.0165 & -0.0009\\ -0.0013 & 0.0009 & -0.0052 & 0.9947 \end{array}\right]$	0.0212	$m_{13}$ $m_{23}$ $m_{32}$
Polarizer 90°	$\left[\begin{array}{cccc} 1.0000 & -0.9782 & -0.0540 & 0.0014\\ -0.9786 & 0.9767 & 0.0536 & -0.0012\\ 0.0727 & -0.0725 & -0.0242 & -0.0004\\ -0.0010 & 0.0015 & -0.0042 & 0.0040 \end{array}\right]$	0.0223	$m_{13}$ $m_{23}$ $m_{31}$ $m_{32}$

**Table 4 sensors-22-02155-t004:** Mueller matrix of a mirror in reflection mode.

Sample	Experimental Mueller Matrix	Mean Error (e)
Mirror	$\left[\begin{array}{cccc} 1.0000 & 0 & 0.0013 & 0.0037\\ 0 & 1.0000 & -0.0009 & 0.0028\\ 0.0005 & 0.0001 & -1.0000 & -0.0003\\ 0.0065 & 0.0018 & 0.0012 & -0.9702 \end{array}\right]$	0.0031

**Table 5 sensors-22-02155-t005:** Mueller matrix of a polarizer with the transmission axis oriented at 90° and aluminum surface in reflection mode.

Sample	Experimental Mueller Matrix
Polarizer 90°	$\left[\begin{array}{cccc} 0.9644 & -0.0228 & 0.1286 & 0.0008\\ -0.0119 & 0.9529 & -0.2915 & 0.0910\\ -0.1615 & 0.0972 & -0.9749 & 0.0136\\ 0.0260 & 0.0650 & 0.0502 & -0.9463 \end{array}\right]$
Aluminum surface	$\left[\begin{array}{cccc} 1.0079 & 0.0073 & 0.1743 & 0.0052\\ -0.0082 & 0.9808 & -0.3409 & -0.0019\\ 0.1497 & 0.1023 & -0.9799 & -0.0011\\ -0.0022 & -0.0018 & 0.0014 & -0.9736 \end{array}\right]$

**Table 6 sensors-22-02155-t006:** Reflection Mueller matrices in fruit surfaces.

Sample	Experimental Mueller Matrix
Apple surface	$\left[\begin{array}{cccc} 0.9912 & -0.0076 & -0.0106 & 0.0012\\ -0.0391 & 0.9301 & -0.6605 & -0.0020\\ 0.1022 & 0.0972 & -0.9749 & 0.0136\\ -0.0020 & 0.0002 & 0.0030 & -0.3438 \end{array}\right]$
Banana surface	$\left[\begin{array}{cccc} 0.9451 & -0.0418 & 0.0025 & -0.0008\\ -0.0691 & 0.8942 & -0.6369 & 0.0025\\ -0.1848 & 0.0212 & -0.6108 & 0.0003\\ -0.0001 & -0.0002 & 0.0030 & -0.1874 \end{array}\right]$

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
