# Peer review of "Stokes Dynamic Polarimeter for Non-Organic and Organic Samples Characterization"

_sensors, 2022, doi:10.3390/s22062155_

Round 1
Reviewer 1 Report
The presented manuscript proposes a Stokes polarimeter system that calculates the Mueller matrix of organic and inorganic samples by the use of transmission or reflection. The Authors demonstrate the whole conception of the experimental setup, with details and the general idea for the flow of the data. Authors propose numerically calculating each element of the matrix by acquiring measured signals and using the 36 measurements method. The results could be interesting for the readers of Sensors, thus can be accepted after minor revision.
- please re-think the title. Maybe "non-organic and organic samples" or "biological and non-biological samples" would sound better?
- Figures 5 and 7 - the quality has to be improved. Do not photograph the oscilloscope, use digital oscilloscope and save the screen-shot as a JPEG. I think that putting the scale on the images would help.
- Please make the Figure 8 and 9, 10, 11 more readable and improve the quality.
- Please re-read the whole manuscript, check all typos. For instance, in line 282 Authors mention temperature but the is no "Celsius" sign.
- the results should be more elaborated in the "Discussion" especially in relation to literature.
Author Response
Response to Reviewer 1 Comments
The presented manuscript proposes a Stokes polarimeter system that calculates the Mueller matrix of organic and inorganic samples by the use of transmission or reflection. The Authors demonstrate the whole conception of the experimental setup, with details and the general idea for the flow of the data. Authors propose numerically calculating each element of the matrix by acquiring measured signals and using the 36 measurements method. The results could be interesting for the readers of Sensors, thus can be accepted after minor revision.
- please re-think the title. Maybe "non-organic and organic samples" or "biological and non-biological samples" would sound better?
Response 1: Thank you for your suggestion. This comment is valuable to explain the objective of our manuscript; “non-organic and organic samples” sounds better for us. As the submission process has been started, we will ask the editor and the assistant editor about the possibility of changing the title at this stage of the submission process.
- Figures 5 and 7 - the quality has to be improved. Do not photograph the oscilloscope, use digital oscilloscope and save the screen-shot as a JPEG. I think that putting the scale on the images would help.
Response 2: Thank you for pointing this out, we totally agree with your comment. We have improved the quality of Figures 5 and 7 in the updated version of the manuscript.
- Please make the Figure 8 and 9, 10, 11 more readable and improve the quality.
Response 3: We have improved the quality of Figures 8 to 11 in the updated version of the manuscript.
- Please re-read the whole manuscript, check all typos. For instance, in line 282 Authors mention temperature but the is no "Celsius" sign.
Response 4: Thanks for your careful review. We have revised the grammar, typos and polished the language of the paper, and included the Celsius unit.
- the results should be more elaborated in the "Discussion" especially in relation to literature.
Response 5: We totally agree with your comment about the Discussion. We included related information to the Discussion sections regarding your recommendation. Please refer to this section displayed in blue text in the revised version of the manuscript (lines 429-445).
“Overall, the Mueller matrices obtained for different elements provide a reliable calibration for ideal reference samples and show that our instrumented array can be configured and adapted to measure lighting properties of additional organic and non-organic materials. Related works proposed Mueller Matrices measured using a CCD camera with higher error accuracy values than our proposed optical array. For instance, in [30], the Mueller Matrices of the air or polarizers using a CCD camera show around 4% of error and 2% in our case. In contrast, reference [31] also measures the Mueller Matrix of the air achieving an error of 0.2% and even lower for some individual elements of the matrix. Both works measure the optical properties of the air in transmission mode but using two different models of CCD cameras, the camera with a higher dynamic range provides notably more accurate results. In our case, we propose utilizing a photodetector instead of a CCD camera because we can vary the acquisition rate of the photodetector and acquire a high number of samples to calculate the Mueller Matrix. Also, the photodetector measurements require fewer saving storage and memory resources. Moreover, this device pro-vides high bandwidth for measuring the polarization properties of the samples under tests. So far, the photodetector offers an alternative to the CCD camera commonly used for Mueller matrix determination.”

Reviewer 2 Report
This article is quite interesting from a metrological point ofview and the accuracy obtained by the authors of the article is quite large.
It would be interesting to look at the accuracy of the results and the errors
if using a CCD camera as a detector.
Author Response
Response to Reviewer 2 Comments
This article is quite interesting from a metrological point of
view and the accuracy obtained by the authors of the article is quite large.
It would be interesting to look at the accuracy of the results and the errors
if using a CCD camera as a detector.
Response: First, thank you for your encouragement to this work. The use of a CCD camera is restrained by the dynamic range, resolution, speed, and bandwidth of the device. In the state of the art, there are Mueller Matrices measured using a CCD camera with higher error accuracy values than our proposed optical array. For instance, the reference [27] measures the Mueller Matrices of the air or polarizers using a CCD camera obtaining around 4% of error, being 2% in our case. In contrast, the reference [28] also measures the Mueller Matrix of the air achieving an error of 0.2%. Both works measure the optical properties of the air in transmission mode but using two different models of CCD cameras, the camera with a higher dynamic range provides notably more accurate results. In our case, we propose the use of a photodetector instead of a CCD camera because we can vary the acquisition rate of the photodetector and acquire a high number of samples to calculate the Mueller Matrix. Also, the photodetector measurements require fewer saving storage and memory resources, moreover, this device provides high bandwidth for measuring polarization properties of the samples under tests. The comparison with the CCD camera array proposed in references [27] and [28] was included in blue font in the Discussion section.

Reviewer 3 Report
In this work, “Stokes dynamic polarimeter for no organic and organic samples characterization”, the authors report an implementation of an automatic Stokes dynamic polarimeter to characterize non-biological and biological material samples. Experiments show the efficiency of the utilized array to calculate the Mueller matrix in reflection and transmission mode for different samples. In addition, the mean square error is calculated for each element of the obtained matrix using referenced values of air and a mirror. Overall, this manuscript has a strong potential for a second review after applying the issues and addressing the shortcomings listed below:
1-The authors should polish/revise some grammatical mistakes and typos along the manuscript. I invite the authors to read their manuscript carefully and make the required changes where necessary.
2-In the Introduction section, while discussing recent developments in the field of biosensing, the following works should be considered and cited to give a more general view to the possible readers of the work: [(i) Photonic and plasmonic metasensors, Laser & Photonics Reviews 16, 2100328 (2021); (ii) Gold nanoparticle based plasmonic sensing for the detection of SARS-CoV-2 nucleocapsid proteins, Biosensors and Bioelectronics 195, 113669 (2022)].
3-In Figures 1-4, use the same font size and font style. Please revise those figures accordingly. Do the same for the remaining figures along the manuscript where necessary.
4-Corresponding references for the equations should be provided if those equations are taken from some other study.
5-Any plans to use the reported implementations for broader biosensing purposes, especially in mid-IR and THz regimes? Please explain. What could be the positive and/or negative sides of the reported implementation for the above mentioned regions?
Author Response
Response to Reviewer 3 Comments
In this work, “Stokes dynamic polarimeter for no organic and organic samples characterization”, the authors report an implementation of an automatic Stokes dynamic polarimeter to characterize non-biological and biological material samples. Experiments show the efficiency of the utilized array to calculate the Mueller matrix in reflection and transmission mode for different samples. In addition, the mean square error is calculated for each element of the obtained matrix using referenced values of air and a mirror. Overall, this manuscript has a strong potential for a second review after applying the issues and addressing the shortcomings listed below:
1.- The authors should polish/revise some grammatical mistakes and typos along the manuscript. I invite the authors to read their manuscript carefully and make the required changes where necessary.
Response 1: Thank you, this comment is valuable. We have revised the grammar, typos and polished the language of the paper. The corresponding changes to address this comment are displayed in blue text along the document.
2.- In the Introduction section, while discussing recent developments in the field of biosensing, the following works should be considered and cited to give a more general view to the possible readers of the work: [(i) Photonic and plasmonic metasensors, Laser & Photonics Reviews 16, 2100328 (2021); (ii) Gold nanoparticle based plasmonic sensing for the detection of SARS-CoV-2 nucleocapsid proteins, Biosensors and Bioelectronics 195, 113669 (2022)].
Response 2: Thank you for your comment. We have included suggested literature. The following text has been added in the Introduction (lines 30-34):
“The design of biosensors has recently shown a considerable advance in human diagnosis through proteins and enzymes detection that characterize bacteria and virus diseases [1]. In detecting specific viruses such as SARS-CoV 2 [2] and more biomedical and clinical diagnoses [3], some of these biosensors use the polarization of light as a detection technique.”
3.- In Figures 1-4, use the same font size and font style. Please revise those figures accordingly. Do the same for the remaining figures along the manuscript where necessary.
Response 3: Thank you for pointing this out. We have re-edited Figures 1-4, and 6. New figures show the title in blue color in the revised version.
4.- Corresponding references for the equations should be provided if those equations are taken from some other study.
Response 4: We have included the reference numbers in the updated version. For eq. 2, reference [22] has been added at line 146.
5.- Any plans to use the reported implementations for broader biosensing purposes, especially in mid-IR and THz regimes? Please explain.
Response 5: As future work, we propose the implementation of a compact and embedded prototype for measuring different kinds of samples. So far, our system is based on Angle Resolver Scatterometer, we expect to characterize and validate the first prototype for selected samples. For future work, we will extend the wavelength in our system to measure and analyze additional properties of the Mueller Matrix elements, not only fundamental polarization elements applied to additional biological samples.
6.- What could be the positive and/or negative sides of the reported implementation for the above mentioned regions?
Response 6: Thank you for this valuable comment. The positive effects could be to include more essential biological tissues to help the diagnosis of abnormalities or cancer. The elements of the Mueller Matrix represent the transformation of the light properties produced by the sample. Thus, extending the wavelength in our system will increase the sensitivity allowing us to measure minimal microstructural changes that characterize most of the biological samples in the Mueller Matrix. The negative side of including mid-IR and THz could be redesign the embedded system for measuring Mueller Matrices due to higher frequencies will be used. However, the Photodetector considered for this work shows the spectral responsivity illustrated in the following plot (extracted from datasheet), therefore, we could still use this device and only change the strategy for acquiring measurements and determine de Mueller Matrices.
Figure 1

Reviewer 4 Report
The authors have proposed an automatic stokes dynamic polarimeter for biological and non-biological samples through Matrix Mueller in reflection and transmission modes.
The Mueller matrix can extract the following parameters:
Isotropic ellipsometry parameters: Ψ and Δ the classical
ellipsometric angles, anisotropic ellipsometry parameters, reflectance, s- and p- reflectance, and depolarization effects in reflection mode.
And Retardance magnitude and orientation, optical rotation / circular retardance, Polarizer transmission axis orientation, circular dichroism- and p- transmittance, polarization-dependent loss, percent transmittance/Insertion loss, and depolarization effects in transmission mode.
But the authors have only represented Mean Error and the elements with higher error. The authors need to mention the parameters which are extracted from the Mueller matrix in section 4 and clarify why only them with the relevant explanation.
The authors mentioned that "Each Mueller matrix obtained is compared with its corresponding reference matrix proposed in the literature [12,13] " in line 297-298, in which [12,13] references are from years (1987,1985) that are way more oldest. The authors should compare with the recent works for a fair comparison.
Diagrams that are not clear (e.g.,fig.5,6,7) need to replace with clear ones and check all the diagrams for proper clearance and visibility to readers.
All tables (table 2 format is odd) should use the same notation and follow one format in the whole manuscript.
Mueller matrix significance needs to be added in Section 2 for the reader's interest.
The authors need to revise the manuscript according to the comments mentioned above.
Author Response
Response to Reviewer 4 Comments
The authors have proposed an automatic stokes dynamic polarimeter for biological and non-biological samples through Matrix Mueller in reflection and transmission modes.
- The Mueller matrix can extract the following parameters:
Isotropic ellipsometry parameters: Ψ and Δ the classical ellipsometric angles, anisotropic ellipsometry parameters, reflectance, s- and p- reflectance, and depolarization effects in reflection mode.
And Retardance magnitude and orientation, optical rotation / circular retardance, Polarizer transmission axis orientation, circular dichroism- and p- transmittance, polarization-dependent loss, percent transmittance/Insertion loss, and depolarization effects in transmission mode.
But the authors have only represented Mean Error and the elements with higher error. The authors need to mention the parameters which are extracted from the Mueller matrix in section 4 and clarify why only them with the relevant explanation.
Response 1: Thank you, this comment is valuable. So far, our dynamic polarimeter is in experimental stage. In this work, our goal is the design and implementation of a reliable system. Our instrumented system requires a calibration state; therefore we highlight only the elements of the Mueller matrices that shows higher errors with respect to ideal Mueller Matrix values proposed to demonstrate the reliability of our polarimetric measurements. Notably, the non-highlighted elements show low accuracy errors which represent high accuracy computation. Further properties and parameters can be derived from Mueller Matrix elements, performing a decomposition or Transformation of the Mueller Matrix elements. These methods are related to the micro structural analysis to characterize biological samples. However, the computation of micro-structural features or the analysis of different organic samples, is reserved to future works.
To address your comment, we have included the following text in the revised version (lines 326-333):
“The following subsections describe the Mueller matrix measured by our proposed dynamic polarimeter and compare the matrix elements that show the higher accuracy error. The comparison results demonstrate the reliability of the instrumented system capable of measuring linear and circular polarization effects of some organic and non-organic samples without using CCD sensors. Additional methods, like Mueller Matrix transformation or decomposition, provide a set of microstructural properties of most biological tissues; however, the comparison and computation of such properties are reserved for future works.”
And lines 341-343:
“The remaining elements of the matrix show a minimal accuracy error regarding ideal values. This comparison indicates that such elements must be considered the most critical during Matrix Mueller calibration due to the higher accuracy errors.”
- The authors mentioned that "Each Mueller matrix obtained is compared with its corresponding reference matrix proposed in the literature [12,13] " in line 297-298, in which [12,13] references are from years (1987,1985) that are way more oldest. The authors should compare with the recent works for a fair comparison.
Response 2: Thank you for your careful review. These matrices are theoretical or ideal, older or recent literature provides the same matrices for reference. To address your comment, we refer these matrices to reference [18] that also describes the ideal values of Mueller Matrices for air, polarimeters and common samples used as reference.
- Diagrams that are not clear (e.g.,fig.5,6,7) need to replace with clear ones and check all the diagrams for proper clearance and visibility to readers.
Response 3: We have improved the quality of Figures 5 to 11 in the updated version of the manuscript.
- All tables (table 2 format is odd) should use the same notation and follow one format in the whole manuscript.
Response 4: Thank you for your comment. Table 2 has been changed to show a similar format in the updated manuscript.
- Mueller matrix significance needs to be added in Section 2 for the reader's interest.
Response 5: Thank you for pointing out this omission. We agree with you regarding the missed information about Mueller matrix significance. To address your comment, we included the following paragraph in section 2 (lines 86-92 highlighted in blue text):
“Mueller Matrix plays an essential role in characterizing the material samples' micro-structural and physical properties. According to the polarization states, different interactions occur between the incident light and the material sample. The individual elements of the Mueller Matrix register the incident beam changes during the light-sample interaction. Hence, calculating such numerical elements in the matrix requires the study of essential concepts about the polarized light generation and the Stokes formalism, which will be briefly presented in the following subsections.”
And a second paragraph in lines 112-117 highlighted in blue text:
“Recently, light polarimetry contributed to analyzing and diagnosing biological tissues, revealing essential information about healthy or pathological tissue status [21]. The structural features in organic or non-organic samples alter the incident light in the sample surface. The changes produced on the incident light after interacting with a material sample can be described by the elements of the Mueller Matrix. These elements represent the coefficients of the incident and output Stokes vector.”
- The authors need to revise the manuscript according to the comments mentioned above.
Response 6: Thank you for your careful review. To address your comment, we have made major updates and changes to the document. The Background section has been updated to include the importance of the Mueller matrix. All Tables have been careful checked for showing similar format. Figures and diagrams have been considerably improved. The experimental results have been rewritten regarding the relevance of all the parameters calculated in the Mueller matrix. The references for comparing Mueller matrix have been updated. Please see all the modifications displayed in blue text in the new version of the manuscript.

Round 2
Reviewer 3 Report
In its current form, the revised manuscript is suitable for publication.
Author Response
Thank you for reviewing our work.

Reviewer 4 Report
From the updated version provided by authors, its been observed that they have made revisions according to the comments highlighted earlier.
I am recommending the article towards the acceptance in this journal if editor agrees.
Author Response
Thank you for reviewing our work.
